# Regulatory network-based imputation of dropouts in single-cell RNA sequencing data

**Ana Carolina Leote**[1,2☯], **Xiaohui Wu**[1,3,4☯], **Andreas Beyer**[1,2,5,6]*

**1** Cluster of Excellence Cellular Stress Responses in Aging-associated Diseases (CECAD), Cologne, Germany, **2** University of Cologne, Faculty of Medicine and Cologne University Hospital, Cologne, Germany, **3** Department of Automation, Xiamen University, Xiamen, China, **4** Pasteurien College, Soochow University, Suzhou, China, **5** Center for Molecular Medicine Cologne (CMMC), Cologne, Germany, **6** Cologne School for Computational Biology & Center for Data Science and Simulation, University of Cologne, Cologne, Germany

☯ These authors contributed equally to this work.
* andreas.beyer@uni-koeln.de

**Data Availability Statement:** The data used in this study are publicly available, as described in the Methods section. Human embryonic stem cell differentiation data are available in GEO under the accession number GSE75748. Renal cell

## Abstract

Single-cell RNA sequencing (scRNA-seq) methods are typically unable to quantify the expression levels of all genes in a cell, creating a need for the computational prediction of missing values ('dropout imputation'). Most existing dropout imputation methods are limited in the sense that they exclusively use the scRNA-seq dataset at hand and do not exploit external gene-gene relationship information. Further, it is unknown if all genes equally benefit from imputation or which imputation method works best for a given gene. Here, we show that a transcriptional regulatory network learned from external, independent gene expression data improves dropout imputation. Using a variety of human scRNA-seq datasets we demonstrate that our network-based approach outperforms published state-of-the-art methods. The network-based approach performs particularly well for lowly expressed genes, including cell-type-specific transcriptional regulators. Further, the cell-to-cell variation of 11.3% to 48.8% of the genes could not be adequately imputed by any of the methods that we tested. In those cases gene expression levels were best predicted by the mean expression across all cells, i.e. assuming no measurable expression variation between cells. These findings suggest that different imputation methods are optimal for different genes. We thus implemented an R-package called ADImpute (available via Bioconductor https://bioconductor.org/packages/release/bioc/html/ADImpute.html) that automatically determines the best imputation method for each gene in a dataset. Our work represents a paradigm shift by demonstrating that there is no single best imputation method. Instead, we propose that imputation should maximally exploit external information and be adapted to gene-specific features, such as expression level and expression variation across cells.

## Author summary

Single-cell RNA-sequencing (scRNA-seq) allows for gene expression to be quantified in individual cells and thus plays a critical role in revealing differences between cells within

carcinoma data are available in the Single Cell Portal (https://singlecell.broadinstitute.org/single_cell/study/SCP1288/tumor-and-immune-reprogramming-during-immunotherapy-in-advanced-renal-cell-carcinoma). Human embryonic kidney (HEK) cell data are available in ArrayExpress (accession code E-MTAB-8735). Oligodendroglioma data are available in GEO under accession number GSE70630. Lung Atlas 10X and Smart-seq2 data are available in Synapse (ID syn21041850). The ADImpute R package, including the transcriptional regulatory network used in this study, is available via Bioconductor: https://bioconductor.org/packages/release/bioc/html/ADImpute.html. All other data are within the manuscript and its Supporting Information files.

**Funding:** A.C.L. received support by the Cologne Graduate School of Ageing Research, funded by the Deutsche Forschungsgemeinschaft (DFG), German Research Foundation, under Germany's Excellence Strategy - EXC 2030/1 - 390661388. X. W. received financial support from the National Natural Science Foundation of China (61871463) and Natural Science Foundation of Fujian Province of China (2017J01068). The funders had no role in study design, data collection and analysis, decision to publish, or preparation of the manuscript.

**Competing interests:** The authors have declared that no competing interests exist.

tissues and characterizing them in healthy and pathological conditions. Because scRNA-seq captures the RNA content of individual cells, lowly expressed genes, for which few RNA molecules are present in the cell, are easily missed. These events are called 'dropouts' and considerably hinder analysis of the resulting data. In this work, we propose to make use of gene-gene relationships, learnt from external and more complete datasets, to estimate the true expression of genes that could not be quantified in a given cell. We show that this approach generally outperforms previously published methods, but also that different genes are better estimated with different methods. To allow the community to use our proposed method and combine it with existing ones, we created the R package ADImpute, available through Bioconductor.

## Introduction

Single-cell RNA sequencing (scRNA-seq) has become a routine method, revolutionizing our understanding of biological processes as diverse as tumor evolution, embryonic development, and ageing. However, current technologies still suffer from the problem that large numbers of genes remain undetected in single cells, although they actually are expressed (dropout events). Although dropouts are enriched among lowly expressed genes, relatively highly expressed genes can be affected as well. Of course, the dropout rate is also dependent on the sampling depth, i.e. the number of reads or transcript molecules (determined with unique molecular identifiers, UMIs) quantified in a given cell. Imputing dropouts is necessary for fully resolving the molecular state of the given cell at the time of the measurement. In particular, genes with regulatory functions—e.g. transcription factors, kinases, regulatory non-coding RNAs (ncRNAs)—are typically lowly expressed and hence particularly prone to be missed in scRNA-seq experiments. This poses problems for the interpretation of the experiments if one aims at understanding the regulatory processes responsible for the transcriptional makeup of the given cell.

The task of correctly imputing dropouts is further complicated by the fact that not all undetected genes are undetected for the same reasons. Some genes are originally expressed in the cell but fail to be detected due to incomplete RNA capturing. These are commonly referred to as technical dropouts. However, some genes are originally not expressed in the cell, and thus not detected (biological zeros). Biological zeros carry information about cell types and states, and incorrect estimation of non-zero expression in these cases may confound cellular profiles [1,2]. Thus, computational methods for dropout imputation face two distinct challenges: on the one hand, to correctly call technical dropouts and, on the other hand, estimate their expression level. If not done carefully, dropout imputation can introduce false positive results in downstream analyses and amplify confounding signals such as batch effects [3].

Most dropout imputation methods are based on the underlying (explicit or implicit) assumption that detected and undetected genes are subject to the same regulatory processes, and hence detected genes can serve as a kind of 'fingerprint' of the state at which the cell was at the time of lysis. Several popular methods are based on some type of grouping (clustering) of cells based on the similarity of their expression patterns. Missing values are then imputed as a (weighted) average across those similar cells where the respective gene was detected [4–7]. For example, the Markov Affinity-based Graph Imputation of Cells (MAGIC) algorithm [4] creates a network of cells by linking cells with similar gene expression signatures. Missing values are subsequently imputed by computing an average over linked cells, where cells get weighted based on how similar or dissimilar their expression signatures are compared to the target cell.

DrImpute [6], scImpute [5] and k-nearest neighbor smoothing (kNN-smoothing) [8] have further developed this notion and have been shown to outperform MAGIC in recent comparisons [3]. These methods rest on two important assumptions: (1) the global expression pattern of a cell (i.e. across the subset of detected genes) is predictive for all genes; (2) the (weighted) average of co-clustering (i.e. similar) cells is a good estimator of the missing value. The first assumption is violated if the expression of a dropout gene is driven by only a small subset of genes and hence the global expression pattern does not accurately reflect the state of the relevant sub-network. Any global similarity measure of the whole transcriptome will be dominated by the majority of genes [3]. The second assumption is violated if the data is scarce, i.e. when either only few similar cells were measured or if the particular gene was detected in only a small subset of cells. In that case the average is computed across a relatively small number of observations and hence unstable.

Methods like Single-cell Analysis Via Expression Recovery (SAVER) [9], or Sparse Gene Graph of Smooth Signals (G2S3) [10], employ a different strategy that can overcome some of these limitations. Instead of using the whole transcriptome of a cell to predict the expression level of a given gene, these methods learn gene-gene relationships from the dataset and use only the specific subset of genes that are expected to be predictive for the particular gene at hand. For example, SAVER learns gene-gene relationships using a penalized regression model, whereas G2S3 optimizes a sparse gene graph. However, if the scRNA-seq dataset at hand is sparse, the usefulness of the gene-gene relationships learnt from that dataset can be limited.

Single Cell RNA-Seq imputAtion constrained By BuLk RNAsEq data(SCRABBLE) [11] is different compared to all of the other methods mentioned above, because it can use bulk sequencing data to assist in the imputation. SCRABBLE combines a de-noising step with a moderated imputation moving the sample means towards the observed (bulk-derived) mean expression values. Limitations of this approach are that first, a matching bulk RNA sequencing data set needs to be available and second that the method only uses external data to adapt the distribution of the single cell data, but does not use it to inform gene-gene relationships.

Here, we compare published approaches that are representative for current state-of-the-art methods to two fundamentally different approaches. The first is a very simple baseline method that we use as a reference approach: we estimate missing values as the average of the expression level of the given gene across all cells in the dataset where the respective gene was detected. Initially intended to serve just as a reference for minimal expected performance, this sample-wide averaging approach turned out to perform surprisingly well and in many instances even better than state-of-the-art methods. The simple explanation is that estimating the average using all cells is a much more robust estimator of the true mean than using only a small set of similar cells, especially when the gene was detected in only few cells and/or if the gene's expression does not vary much across cells.

The second new approach avoids using a global similarity measure comparing entire transcriptomes. Instead, similar to SAVER or G2S3 it rests on the notion that genes are part of regulatory networks and only a small set of correlated or functionally associated genes should be used to predict the state of undetected genes. However, unlike other methods, we propose to use transcriptional regulatory networks trained on independent (bulk seq) data to rigorously quantify the transcriptional relationships between genes. Missing values are then imputed using the expression states of linked genes in the transcriptional regulatory network and exploiting the known quantitative relationships between genes. This approach allows imputing missing states of genes even in cases where the respective gene was not detected in any cell or in only extremely few cells. This second new approach rests on the assumption that the network describes the true regulatory relationships in the cells at hand with sufficient accuracy. Here, we show that this is indeed the case and that combining the two new approaches with

published state-of-the-art methods drastically improves the imputation of scRNA-seq dropouts. Importantly, the performance of an imputation method is dependent on the 'character' of a gene (e.g. its expression level or the variability of expression between cells). Hence, we implemented an R-package (Adaptive Dropout Imputer, or ADImpute) that determines the best imputation method for each gene through a cross-validation approach.

## Results

### Imputing dropouts using a transcriptional regulatory network

In order to understand whether the inclusion of external gene regulatory information allows for more accurate scRNA-seq dropout imputation, we derived a regulatory network from bulk gene expression data in 1,376 cancer cell lines with known karyotypes. While the expression levels of genes in this data will be cell type-specific, the relationships between genes (e.g. concerted up-regulation of a transcription factor and its targets) are frequently conserved across cell types, allowing us to pool the data together to learn a generic gene regulatory network. For this purpose, we modelled the change (compared to average across all samples) of each gene as a function of its own copy number state and changes in predictive genes:

$$y_i = \alpha_i \cdot c_i + \sum_{j \neq i} \alpha_{ij} \cdot y_j + \varepsilon_i, \tag{1}$$

where $y_i$ is the expression deviation (log fold change) of gene $i$ from the global average, $c_i$ is the known (measured) copy number state of gene $i$, $\alpha$ the vector of regression coefficients, $y_j$ the observed change in expression of gene $j$ and $\varepsilon_i$ the i.i.d. error of the model. To estimate a set of predictive genes $j$, we made use of Least Absolute Shrinkage and Selection Operator (LASSO) regression [12], which penalizes the L1 norm of the regression coefficients to determine a sparse solution. LASSO was combined with stability selection [13] to further restrict the set of predictive genes to stable variables and to control the false discovery rate (Methods). This approach ensures that the algorithm only selects gene-gene relationships that are invariant across most or all training data. Thus, interactions that would be specific to a single cell type will be excluded from the model. Using the training data, models were fit for 24,641 genes, including 3,696 non-coding genes. The copy number state was only used during the training of the model, since copy number alterations are frequent in cancer and can influence the expression of affected genes. If copy number states are known, they can of course also be used during the dropout imputation phase. Using cell line data for the model training has the advantage that the within-sample heterogeneity is much smaller than in tissue-based samples [14]. However, in order to evaluate the general applicability of the model across a wide range of conditions, we validated its predictive power on a diverse set of tissue-based bulk-seq expression datasets from the The Cancer Genome Atlas (4,548 samples from 13 different cohorts; see Methods and S1 Fig) and the Genotype-Tissue Expression (17,382 samples from 30 different healthy tissues; see Methods and S2 Fig).

Such a model allows us to estimate the expression of a gene that is not quantified in a given cell based on the expression of its predictors in the same cell. Here, the difficulty lies in the fact that imputed dropout genes might themselves be predictors for other dropout genes, i.e. the imputed expression of one gene might depend on the imputed expression of another gene. In order to derive the imputation scheme based on the model from Eq (1), we revert to an algebraic expression of the problem,

$$Y = AY, \tag{2}$$

where $A$ is the adjacency matrix of the transcriptional network, with its entries $\alpha_{ij}$ being fitted

using the regression approach described above, and $Y$ is the vector of gene expression deviations from the mean across all cells in a given cell. In the current implementation we assume no copy number changes and hence, we exclude the $c_i$ term from Eq (1). Like in Eq (1), we omit the intercept since we are predicting the deviation from the mean. Subsequently, imputed values are re-centered using those means to shift imputed values back to the original scale (see Methods). Further note that we drop the error term $\varepsilon$ from Eq (1), because this is now a prediction task (and not a regression). Here, we exclusively aim to predict dropout values, and (unlike SAVER) our goal is not to improve measured gene expression values. Hence, measured values remain unchanged. It is therefore convenient to further split $Y$ into two sub-vectors $Y^m$ and $Y^n$, representing the measured and non-measured expression levels, respectively. Likewise, $A$ is reduced to the rows corresponding to non-measured expression levels and split into $A^m$ (dimensionality $|n|\times|m|$) and $A^n$ (dimensionality $|n|\times|n|$), accounting for the contributions of measured and non-measured genes, respectively. The imputation problem is then reduced to:

$$Y^n = A^n Y^n + A^m Y^m \qquad (3)$$

As $Y^m$ is known (measured) and will not be updated by our imputation procedure, the last term can be condensed in a fixed contribution, $F = A^m Y^m$, accounting for measured predictors:

$$Y^n = A^n Y^n + F \qquad (4)$$

The solution $Y^n$ for this problem is given by:

$$Y^n = (I - A^n)^{-1} F \qquad (5)$$

The matrix $(I-A^n)$ may not be invertible, or if it is invertible, the inverse may be unstable. Therefore, we computed the pseudoinverse $(I-A^n)^+$ using the Moore-Penrose inversion. Computing this pseudoinverse for every cell is a computationally expensive operation. Thus, we implemented an additional algorithm finding a solution in an iterative manner (Methods). Although this iterative second approach is not guaranteed to converge, it did work well in practice (see S3 Fig, Methods). While our R-package implements both approaches, subsequent results are based on the iterative procedure.

## Transcriptional regulatory network information improves scRNA-seq dropout imputation

To assess the performance of our network-based imputation method and compare it to that of previously published methods, we considered eight different single-cell RNA-sequencing datasets [15–19], covering a wide range of sequencing techniques (Smart-seq versions 1, 2 and 3 and droplet-based method 10X) and biological contexts (healthy tissue, cancer, stem cell differentiation and Human Embryonic Kidney—HEK—cells). A summary of the dataset characteristics, including number of cells and average number of quantified genes per cell, is provided in S1 Table. It was important to include a range of different healthy cell types in the evaluation, because the transcriptional regulatory network was trained on cancer cell line data. Thus, by including data from non-cancerous tissues, we could evaluate possible restrictions induced by the model training data.

In order to quantify the performance of both proposed and previously published imputation methods, we randomly set a fraction of the quantified values in the test data to zero according to two different schemes (Methods) and stored the original values for later comparison with the imputed values. Imputation was then performed on the masked dataset using our network-based approach, DrImpute [6], kNN-smoothing [8], SAVER [9], scImpute [5] and

SCRABBLE [11]. Those methods were chosen since they were shown to be among the top-performing state-of-the-art dropout imputation methods [20,21]. For masked entries imputed by all tested methods, the quality of imputation was assessed for each gene, using two approaches: computing the correlation between observed and imputed values for each gene, and computing the Mean Squared Error (MSE) of imputation (Methods).

According to the correlation quality measure our network-based approach (called 'Network') mostly outperformed other methods (Fig 1), especially in UMI-based datasets. We quantified the percentage of genes in the transcriptome of each dataset best imputed by each method (highest correlation) and verified that in six out of our eight test datasets, the network-based approach resulted in the highest performance for most genes (Table 1). Additionally, we observed that Network was less affected by low average expression levels compared to all other imputation methods (Figs 1 and S4, expression quartiles Q1 and Q2). This was expected since our network-based approach relies on information external to the dataset for dropout imputation, while other methods require sufficient observations of a gene to learn its expression characteristics from the single cell data itself. This result is in line with, for instance, a previous observation that scImpute is sensitive to missing information about genes across cells [20]. SCRABBLE is able to incorporate the average expression in matched bulk RNA-seq data to aid imputation, effectively taking advantage of external information like Network. Although this information is not available for most scRNA-seq datasets, it was available for the human embryonic stem cell (hESC) dataset, prompting us to use it in an additional SCRABBLE test (Figs 1, S4 and S5). We observed that SCRABBLE's performance was not improved when incorporating this additional information.

As the network-based approach uses information regarding other genes contained in the same cell, we hypothesized its accuracy might be more affected by increasingly sparse information per cell when compared to other methods. However, the relative performance differences between methods were largely invariant to the number of missing genes per cell (S7 Fig), suggesting that other methods also suffer from the scarcity of information for cells with low sampling efficiency. The sensitivity of the imputation to the proportions of missing values was dependent on a multitude of factors, including the quality of the data, the specific gene(s) expressed in the cell type(s) at hand, the number of missing values, variability of expression/ cell type heterogeneity within the study and possibly many others. Further, the network-based method performed well over a range of different cell types and showed decreased performance upon randomization of the transcriptional network (Methods, S8 Fig). Thus, the diversity of cell lines used in the training data seemed to capture a large fraction of all possible regulatory relationships in the human transcriptome.

We additionally evaluated imputation performance using the mean squared error (MSE) between original and imputed values. In this analysis, we included a 'Baseline' method, which imputes dropouts with the average expression value of the gene across all cells, as a reference. Using the MSE as an alternative metric of performance, we also observed that Network outperformed previously published methods across all tested datasets and expression quartiles (S6 Fig). However, the performance of the Baseline method (S6 Fig, grey violin), which does not account for any expression variation between cells, was surprising to us. While SAVER showed a poor performance in comparison to all other methods (also described in [20]), it should be noted that this method aims to estimate the true value for all genes, not only for the dropout genes. This results in a change of both dropout and quantified values, which explains the good correlation performance but relatively high MSE.

Taken together, these results indicate that Network often–but not always—leads to more accurate imputations than state-of-the-art imputation methods (S6 Fig), while preserving variation across cells as captured by the correlation analysis (Figs 1 and S4).

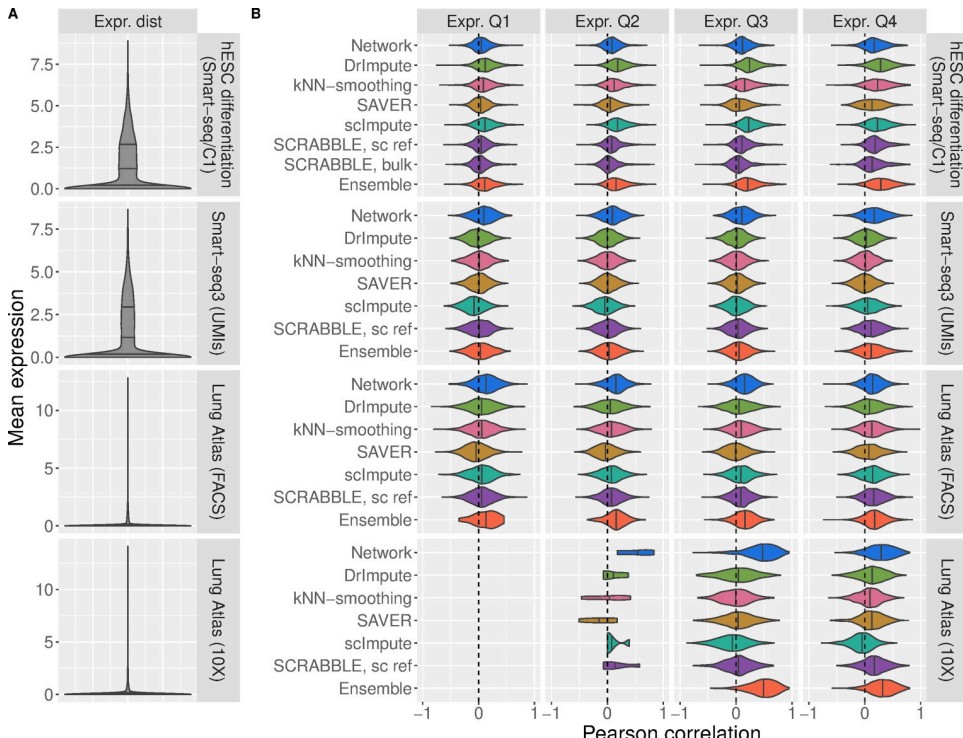

**Fig 1. Imputation performance of Network (blue), DrImpute (green), kNN-smoothing (pink), SAVER (yellow), scImpute (turquoise), SCRABBLE (purple) and Ensemble (orange). A)** Distribution of average expression levels (per gene) in each dataset. Quartiles are represented by vertical lines. **B)** Pearson correlation coefficient, for each gene, between the imputation by the specified method and the original values before masking. Only values that could be imputed by all methods (non-zero imputation) were considered. Correlations per gene across cells were computed for all genes for which at least 10 imputation values were available for analysis. Expression quartiles were determined for each dataset separately, on the masked data.

However, our analysis also uncovered that the advantage of using specific methods varies between datasets and expression quartiles, suggesting that there is no universally best performing method that outperforms the others in all cases. This motivated us to develop an ensemble approach, where we determine in a cross-validation scheme the best imputation method for each gene in the dataset at hand. We tested its performance (Figs 1, S4, and S6) and observed that the ensemble method tends to approach the performance of the best performing method.

**Table 1. Percentage of genes best imputed by each method (highest Pearson correlation coefficient) in the seven test datasets restricted to values that could be imputed by all individual methods.**

|  | Network | DrImpute | kNN-smoothing | SCRABBLE, bulk | SCRABBLE, sc ref | scImpute | SAVER |
|---|---|---|---|---|---|---|---|
| hESC differentiation | 14.0% | **31.0%** | 13.2% | 5.9% | 7.5% | 22.0% | 6.3% |
| hESC time course | 16.6% | **29.0%** | 22.7% | 6.7% | 9.2% | 10.4% | 5.5% |
| Oligodendroglioma | **29.3%** | 18.4% | - | - | 15.0% | 20.1% | 17.1% |
| Lung Atlas (10X) | **58.4%** | 11.7% | 6.0% | - | 14.4% | 2.3% | 7.2% |
| Lung Atlas (FACS) | **31.5%** | 13.5% | 14.7% | - | 17.8% | 15.1% | 7.4% |
| Renal Cell Carcinoma | **63.2%** | 5.3% | 3.1% | - | 12.2% | 2.0% | 14.2% |
| Smart-seq3 (reads) | **26.8%** | 11.3% | 12.1% | - | 18.1% | 19.4% | 12.3% |
| Smart-seq3 (UMIs) | **32.3%** | 12.8% | 12.4% | - | 18.2% | 11.5% | 12.9% |

## Gene features determine the best performing imputation method

To better understand what gene features drive the performance differences between methods, we characterized the genes best imputed by each of the methods. We determined, for each gene in each test dataset, the method resulting in the highest correlation between imputed and original values (Table 1). Methods with performance close to the best (correlation difference of 0.1, see Methods) were considered to be "top performers". We chose this approach because some methods apply similar strategies for dropout imputation and thus are expected to perform best for the same set of genes. Our analysis allows to capture these similarities between methods. The genes for which a given method was among the "top performers" were compared against a background including all genes for which all methods were able to perform imputations (Figs 2 and S10). As expected, methods relying on the similarity of cellular transcriptomes (scImpute, DrImpute and kNN-smoothing) performed best for more frequently detected genes (lower percentage of NAs per gene across cells). Conversely, the Network (and, to some extent, SAVER and SCRABBLE) were among the top performers especially on genes with many missing values. SAVER and Network are model-based methods, not relying on the comparison of entire transcriptomes between cells. SCRABBLE and Network are methods using external information for the imputation. Based on the above results we conclude that both aspects (model-based imputation and using external information) are advantageous for the imputation of rarely detected genes. We also determined the top performing methods for each gene based on the MSE instead of correlations. We observed that, as expected, the Baseline method performed best for genes with low expression levels and low variance (S11 Fig).

## Network-based imputation aids downstream analyses

**Cell trajectory inference.**   scRNA-seq data is particularly suitable for the study of dynamic processes such as development or differentiation, due to the high numbers of individual cells sequenced and differences in progression along the dynamic process of choice between them [22]. Here, we make use of a time course hESC differentiation dataset [16] containing six distinct timepoints to infer cell trajectories through the course of differentiation, in order to assess the impact of dropout imputation on the inferred trajectories. To this end, we computed cell trajectories before and after imputation with slingshot [23], a method well evaluated in an independent work [21], and compared them to the known timepoint labels of the dataset (Fig 3). Imputation with any of the tested methods led to a better agreement between timepoint labels and inferred pseudotime, highlighting the usefulness of dropout imputation for downstream analyses such as trajectory inference. Additionally, Baseline and bulk-based SCRABBLE imputations showed the worst performance among the compared methods. Baseline's poor performance was expected, as signals across the whole dataset were averaged for dropout imputation, which dilutes the progressive changes across the course of differentiation. However, it was surprising to us that using Baseline was still better than not performing any dropout imputation. A possible explanation might be that leaving technical zeros in the data introduces additional noise thereby complicating the correct positioning of cells on the pseudo time trajectory. Bulk-based SCRABBLE's poor performance may be explained by the small number of samples per time point available in the bulk reference (n = 2 or 3). An average across such few samples is unstable, and it remains to be seen whether a more reliable bulk reference results in better performance for SCRABBLE. Finally, our results support the use of an Ensemble method where the best performing method is picked for each gene via a cross-validation approach, as the performance of this Ensemble method was practically indistinguishable from the best performing method.

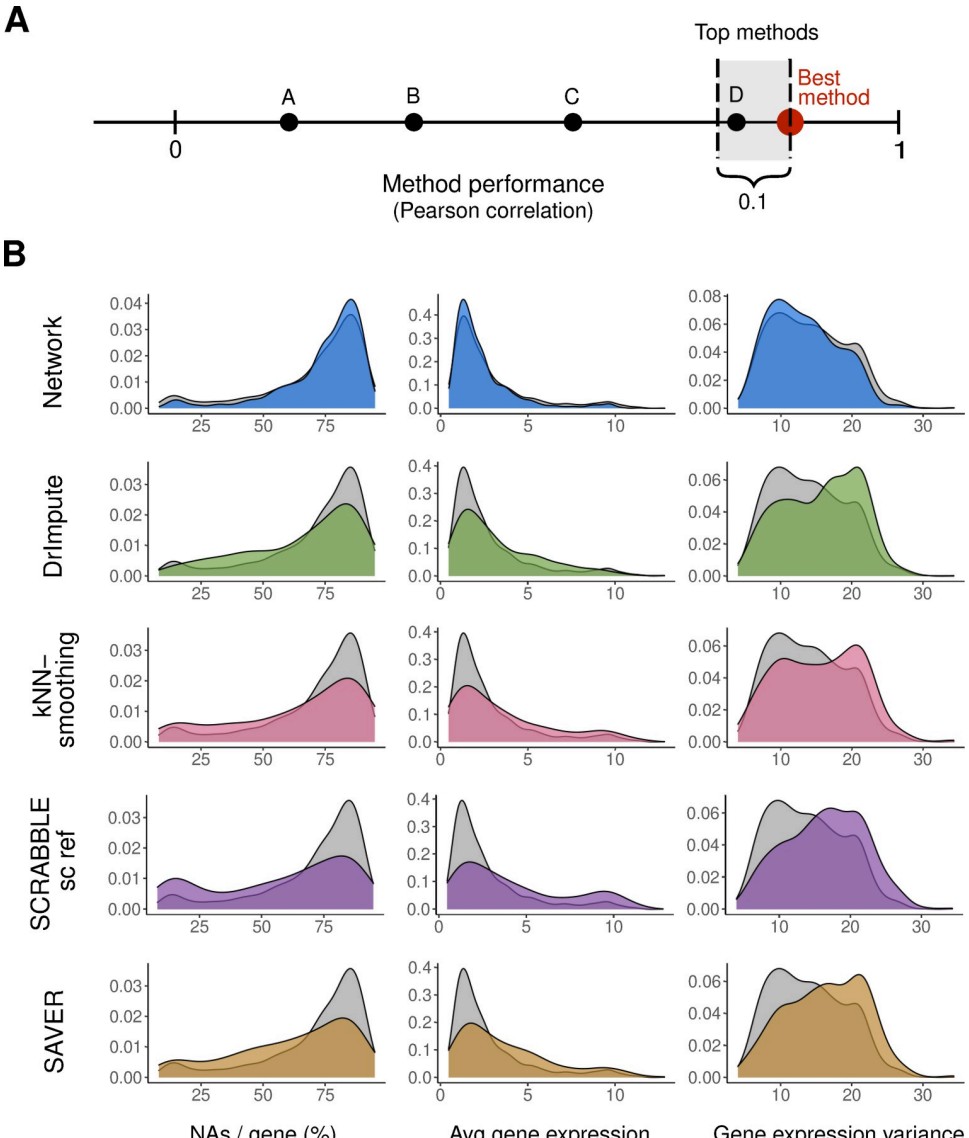

**Fig 2. Characterization of the genes best predicted by Network (blue), DrImpute (green), kNN-smoothing (pink), SCRABBLE (bulk and single cell data as reference; purple) and SAVER (yellow) in the Lung Atlas 10X dataset. A)** Determination of the top best performing methods for each gene. Methods performing best and close to the best method (correlation not smaller than 0.1 –best method) were selected as top performers. Genes for which all methods were top performers were included in the background but not in the foreground. **B)** Distribution of missing values per gene, average expression levels and variance of the genes for which a given method is one of the top performers, compared against all tested genes (background). Average gene expression is shown as $\log_2$-transformed normalized expression. Too few genes were best predicted by scImpute, so no distributions were drawn for this method.

**Data visualization.** Another popular application of scRNA-seq is the identification of discrete sub-populations of cells in a sample in order to, for example, identify new cell types. The clustering of cells and the visual 2D representation of single-cell data is affected by the choice of the dropout imputation method [20]. Therefore, we assessed the impact of dropout imputation on data visualization using Uniform Manifold Approximation and Projection (UMAP) [24] on the hESC data before and after imputation by all methods. The snapshot hESC dataset was particularly suitable in this case, because it was of high quality and it consisted of six well-

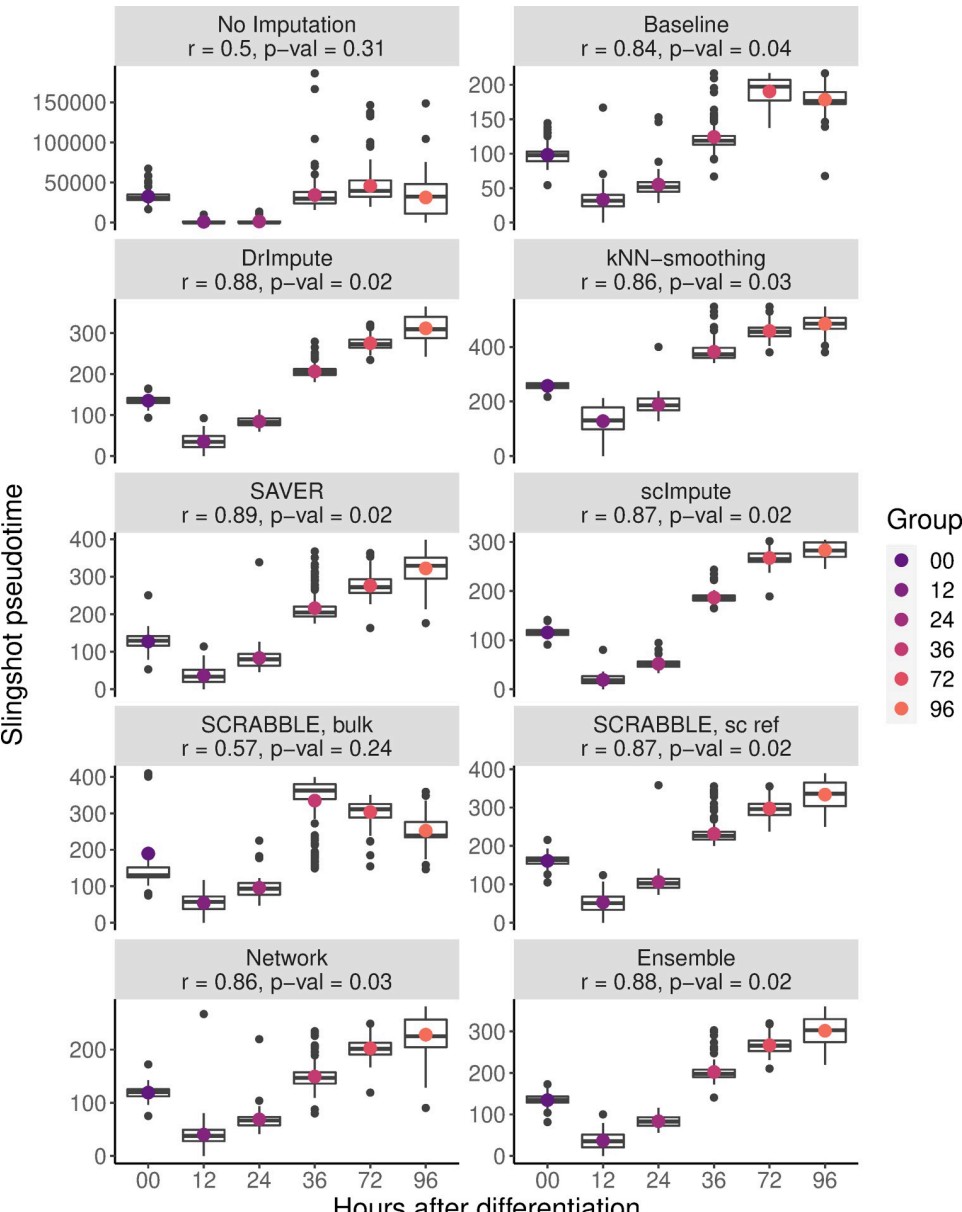

**Fig 3. Comparison of impact of different imputation methods on cell trajectory inference for time course hESC differentiation data.** Cells were projected onto the trajectories determined with slingshot to compute a pseudotime of differentiation. The pseudotime assigned to each individual cell (y-axis) is then compared to the time point label of the sample (x-axis). Colored points represent the mean pseudotime per known time point. In the title, the Pearson's r between pseudotime and time point labels (and the respective p-value) are shown. Note that slingshot always fails to correctly position the sample at time point 0, suggesting an artifact in the original data.

annotated distinct cell types. This analysis confirmed that the choice of the imputation method impacts on the grouping/clustering of cells (S12 Fig). Application of other dimensionality reduction techniques (t-distributed stochastic neighbor embedding, t-SNE, and Zero-Inflated Negative Binomial-based Wanted Variation Extraction ZINB-WaVE) showed varying results depending on the chosen method, suggesting that visual clustering upon dimensionality reduction is an inconclusive criterion for evaluating dropout imputation (S13 Fig).

**Cluster marker detection.** We next asked to what extent the detection of cluster markers would be affected by the choice of the imputation method. Thus, we applied Seurat [25] to the hESC differentiation dataset, which was composed of a well-defined set of distinct cell types, before and after imputation. We then defined genes that were significantly differentially expressed between one cluster and all the others as cluster markers (Methods). We observed a considerable overlap between markers detected before and after applying the tested imputation methods (Fig 4A, horizontal dashed line), suggesting a common core of detected cluster markers across methods. Additionally, the numbers of significant markers detected after Network and Baseline imputations were lower than for other imputation methods (Fig 4A). Imputation with kNN-smoothing, scImpute and, to a smaller extent, with DrImpute, led to the highest number of significant markers (Fig 4A). We hypothesized that many of these marker genes may result from artefactual clustering of cells. In order to test that notion we first determined all Gene Ontology (GO) biological process terms that were enriched in the respective cell clusters without any dropout imputation. We termed them 'high confidence GO terms' since they are independent of the choice of the imputation method. It turned out that kNN-smoothing, scImpute and DrImpute had the weakest enrichments in high confidence GO biological process terms (Fig 4B and 4C; Methods; S4 Table), suggesting that the extra markers found upon applying scImpute and DrImpute contained many false positives, which diluted biological signals. Conversely, Network and SCRABBLE led to the strongest enrichments in high confidence GO biological process terms (Fig 4B and 4C).

**Determining transcriptional regulators.** Genes with regulatory functions are particularly important for understanding and explaining the transcriptional state of a cell. However, since genes with regulatory functions are often lowly expressed [26], they are frequently subject to dropouts. Since our analysis had shown that the network-based approach is especially helpful for lowly expressed genes (Fig 2), we hypothesized that the imputation of transcript levels of regulatory genes would be particularly improved. In order to test this hypothesis, we further characterized those cluster markers that were exclusively detected using the network-based method. Indeed, we observed regulatory genes to be enriched among those markers (Fig 5A). The transcription factor ETS Homologous Factor (EHF) was the second most significant trophoblast-specific among these markers exclusively detected upon network-based imputation. EHF is a known epithelium-specific transcription factor that has been described to control epithelial differentiation [27] and to be expressed in trophoblasts (TB) [28], even though at very low levels (EHF expression was found among the first quintile of bulk TB RNA-seq data from the same authors). While EHF transcripts were not well captured in TB single-cell RNA-seq data (only detected in 39 out of 775 TB cells), a trophoblast-specific expression pattern was recovered after network-based imputation (Fig 5B, upper panel), but not with any of the other tested imputation methods (S14 Fig). Similarly, Odd-Skipped Related Transcription Factor 1 (OSR1) has been described as a relevant fibroblast-specific transcription factor [29] which failed to be detected without imputation. Imputing with Network lead to the strongest fibroblast-specific expression pattern of OSR1 (Figs 5B, lower panel, and S14). Interestingly, TWIST2 and PRRX1, described by Tomaru *et al.* [29] to interact with OSR1, also showed fibroblast-specific expression (S15 Fig). Taken together, these results suggest that imputation based on transcriptional regulatory networks can recover the expression levels of relevant, lowly expressed regulators affected by dropouts.

## Discussion

This work has led to the following key findings: (i) a model-based approach using external data is particularly powerful for the imputation of rarely detected genes; (ii) based on the MSE

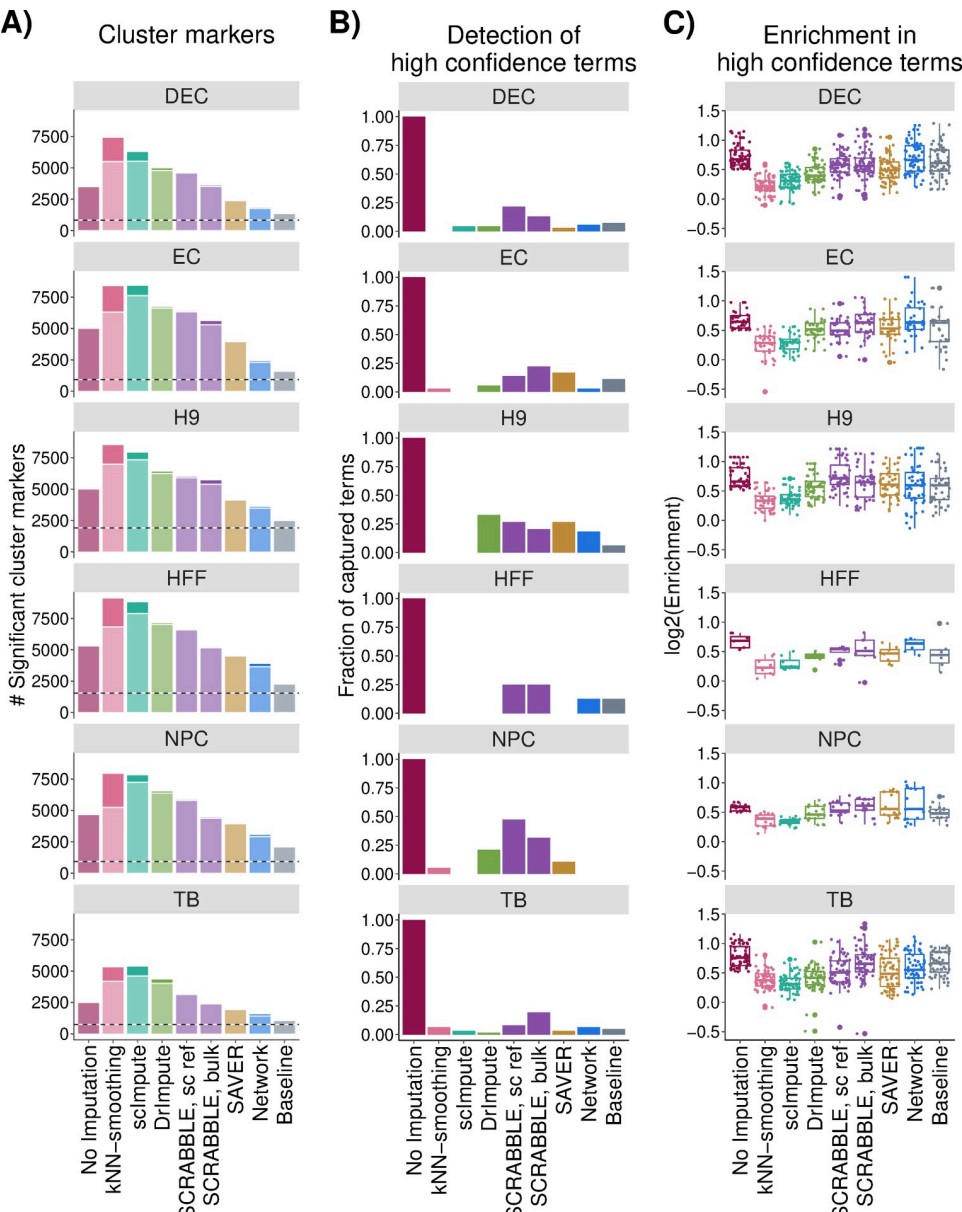

**Fig 4. Detection of cell type-specific markers before and after imputation. A)** Number of significant (FDR < 0.05, |log$_2$FC| > 0.25) cell type markers detected with no dropout imputation and using the tested imputation methods. Horizontal dashed lines correspond to the number of markers detected irrespectively of imputation. The fraction of the bar in a darker shade corresponds to the number of markers detected exclusively when using a given imputation approach. **B)** and **C)** fraction of captured high confidence terms, defined as significantly enriched (p.value < 0.001 and log$_2$Enrichment > 0.5, Methods) GO biological process terms among the cluster markers detected without imputation. **B)** Sensitivity: fraction of high confidence terms detected as significantly enriched (p.value < 0.001 and log$_2$Enrichment > 0.5) among the cluster markers detected with each imputation method. **C)** log$_2$-enrichment of all high confidence terms among the cluster markers detected with each imputation method. DEC: definitive endoderm cells; EC: endothelial cells; H9: undifferentiated human embryonic stem cells; HFF: human foreskin fibroblasts; NPC: neural progenitor cells; TB: trophoblast-like cells.

criterion the expression of surprisingly many genes is best predicted by simply using their average expression across cells ('Baseline'); (iii) not all genes are equally well predicted by a single imputation approach; instead, one should adapt the imputation method to the specific

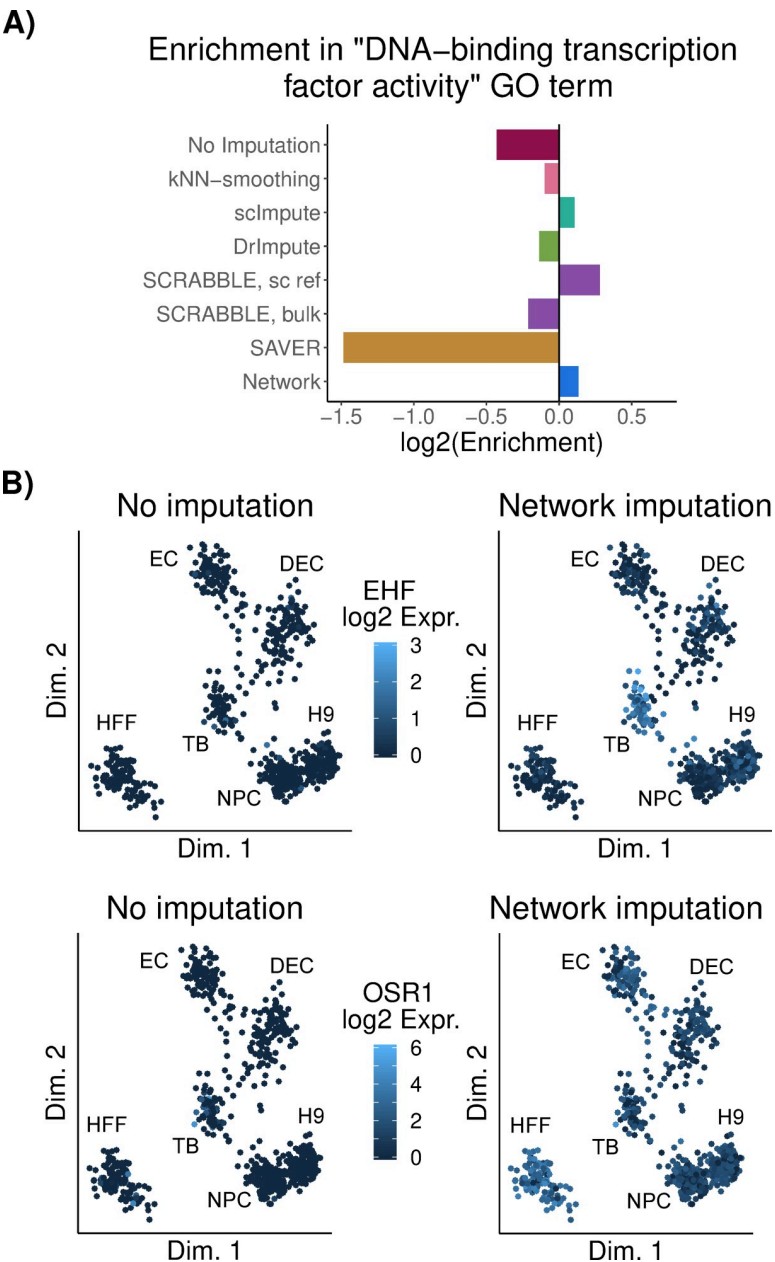

**Fig 5. Detection of cell type-specific transcription factors is improved upon network-based imputation. A)** Enrichment score in GO term "DNA-binding transcription factor activity" among the genes uniquely detected after each imputation approach. **B)** Projection of cells onto a low dimension representation of the data before imputation, using ZINB-WaVE[30]. Color represents normalized expression levels of EHF (top) and OSR1 (bottom) before and after Network-based imputation. DEC: definitive endoderm cells; EC: endothelial cells; H9: undifferentiated human embryonic stem cells; HFF: human foreskin fibroblasts; NPC: neural progenitor cells; TB: trophoblast-like cells.

gene in a given dataset. In addition, our work confirmed earlier findings, such as the artifactual clustering resulting from some imputation methods.

The consideration of external gene co-expression information for the dropout imputation substantially improved the performance in many cases, especially for lowly expressed genes. Since genes with regulatory functions are often lowly expressed [26], imputation of those

genes might be critical for explaining expression variation between cells. Of note, cell type-specific regulatory genes were successfully imputed using information from our global gene co-expression network (Fig 5). This observation, together with the demonstrated predictive capacity of our network across cancer and healthy human data from a wide range of tissues, highlights the transferability of the gene-gene relationships learnt by our network. scImpute, kNN-smoothing and DrImpute elevated the number of cluster markers found in downstream DE analysis (Fig 4), but these additional markers seemed to dilute the true biological signal in the data. A similar behaviour has been described elsewhere for scImpute [3]. Similarly to our network-based approach, SAVER makes use of gene-gene relationships for imputation. However, SAVER learns these relationships in the dataset at hand, while Network makes use of externally trained relationships. The fact that Network outperforms SAVER in our comparisons suggests that the scRNA-seq data at hand may often be too sparse for relationships to be adequately learnt.

A potential limitation of our approach is that a transcriptional network derived from bulk-seq data may not fully capture gene-gene relationships that are detectable from single cell data. For example, gene regulatory relationships that are specific to a small sub-population of cells in a bulk tissue may not be correctly captured, because the signal would be too weak. A second example would be genes regulated during the cell cycle. Bulk tissue is usually not synchronized, i.e. it consists of a mix of cells at different cell cycle stages, which may prevent the detection of those relationships. To some extent these limitations were alleviated by using cell line data rather than actual tissue data for training the network. Of course, the network that we used here is still imperfect. However, despite that imperfection it demonstrated the power of our approach. Using it was clearly advantageous over not using it in most dropout imputation tests, and we showed its predictive power across independent datasets from cancer, healthy human tissue and cell lines.

A surprising finding of our analysis is the fact that the sample-wide average expression ('Baseline') performs well for the imputation of many genes when using the MSE as a performance measure (S3 Table). Originally, we developed Baseline only as a benchmark of a minimalistic approach in order to compare the performance gain of more sophisticated approaches against this one. Much to our surprise, many genes could not be imputed better by any of the competing approaches. As expected, genes whose expression levels were best imputed by Baseline were characterized by lower variance across cells and by remaining undetected in relatively many cells (S11 Fig). A potential problem of methods based on co-clustering cells is that the number of observations per cluster can get very small, which makes the estimation of the true mean more unstable. Thus, averaging across all cells is preferred when the gene was detected in only few cells and/or if the gene's expression does not vary much across cells. Further, our findings imply that cell-to-cell expression variation of many genes is negligible or at least within the limits of technical measurement noise. Obviously, Baseline does not support the identification of differentially expressed genes between different groups of cells. However, it may help reducing artifacts resulting from the clustering of cells with technical zeros (Fig 3).

The third—and maybe most important—conclusion is that the best performing imputation method is gene- and dataset-dependent. That is, there is no single best performing method. If the number of observations is high (many cells with detected expression) and if the expression quantification is sufficiently good, scImpute and DrImpute outperformed other methods. Importantly, the technical quality of the quantification depends on the read counts, which in turn depends on sampling efficiency, gene expression, transcript length and mappability–i.e. multiple factors beyond expression. If however, gene expression is low and/or too imprecise, scImpute and DrImpute were outcompeted by other methods, since expression data across neighboring cells is not informative in this case. This finding led us to conclude that a

combination of imputation methods would be optimal. Hence, we developed an R-package that determines 'on the fly' for each gene the best performing imputation method by masking observed values (i.e. *via* cross validation). This approach has the benefit that it self-adapts to the specificities of the dataset at hand. For example, the network-based approach might perform well in cell types where the assumptions of the co-expression model are fulfilled, whereas it might fail (for the same gene) in other cell types, where these assumptions are not met. Hence, the optimal imputation approach is gene- and dataset-dependent. An adaptive method selection better handles such situations. Another benefit of this approach is that the cross validation imputation performance can be used as a quantitative guide on how 'imputable' a given gene is in a specific scRNA-seq dataset. We have therefore implemented and tested this approach (see Figs 1, S4, and S5). The resulting R-package (called ADImpute) is open to the inclusion of future methods and user-provided gene networks, includes scImpute's estimation of dropout probability and it can be downloaded from https://bioconductor.org/packages/release/bioc/html/ADImpute.html. While, by default, ADImpute makes use of the transcriptional regulatory network described here, the user can provide any desired gene network e.g. for a different species, similarly to [31]. Our network has been trained on a large set of human cancer cell lines and thus is expected to capture much of the possible regulatory interactions of interest to many users; however, ADImpute facilitates the use of custom-made more specialized networks.

We believe that this work presents a paradigm shift in the sense that we should no longer search for the single best imputation approach. Rather, the task for the future will be to find the best method for a particular combination of gene and experimental condition.

## Methods

### Pre-processing of cancer cell line data for transcriptional regulatory network inference

Entrez IDs and corresponding gene symbols were retrieved from the NCBI (https://www.ncbi.nlm.nih.gov/gene/?term=human%5Borgn%5D). Genome annotation was obtained from Ensembl (*Biomart*). Finally, genes of biotype in protein coding, ncRNA, snoRNA, scRNA, snRNA were used for network inference. For CCLE [32], 768 cell lines that were used in Seifert et al. [14] were used. Raw CEL files were downloaded from https://portals.broadinstitute.org/ccle/ and processed using the R package RMA in combination with a BrainArray design file (HGU133Plus2_Hs_ENTREZG_21.0.0). Final expression values were in $\log_2$ scale. Expression levels and CNV data set from RNA-seq were downloaded from Klijn et al. [33]. Before combining, each dataset is $\log_2$ transformed and scaled to (0,1) for all genes in each sample using R function scale. Then datasets were merged and the function ComBat from the sva R package [34] was used to remove batch effect of the data source. The final combined data set contains 24641 genes in 1443 cell lines. Finally, expression levels of genes were subtracted by the average expression level across all cell lines of the corresponding gene.

### Network inference based on stability selection

The network inference problem can be solved by inferring independent gene-specific sub-networks. We used the linear regression model from Eq (1) to model the change in a target gene as dependent on the combination of the gene-specific CNA and changes in all other genes. Here the intercept is not included because the data is assumed to be centered. We used LASSO with stability selection [13] to find optimal model parameters $\alpha_{ij}$.

The R package *stabs* was employed to implement stability selection and the *glmnet* package was used to fit the generalized linear model. Two parameters regarding error bounds were set with the cutoff value being 0.6 and the per-family error rate being 0.05. A set of stable variables were defined by LASSO in combination with stability selection. Then coefficients of the selected variables were estimated by fitting generalized linear models using the R function *glm*.

## Network validation using TCGA and GTEx data

Gene expression and gene copy number data of 14 different tumor cohorts (4548 tumor patients in total) from TCGA collected in a previous study [14] were used for validation. We examined the predictive power of our inferred networks on each TCGA cohort by predicting the expression level of each gene for each tumor using the corresponding copy number and gene expression data.

Additionally, in order to validate the applicability of the learnt network to healthy tissues, we further leveraged gene expression data from the Genotype-Tissue Expression (GTEx) Project. Read counts were downloaded from the portal website (version 8), normalized using the R package *DESeq2* and centered gene-wise across tissues.

For each TCGA cohort or GTEx tissue, the expression levels of each gene were predicted using the network and expression quantification of the interacting genes in the same sample. The predicted value was then compared to the observed value, present in the original dataset. The quality of prediction for each TCGA cohort or GTEx tissue was quantified as either the correlation between predicted and observed expression of a gene across all samples or the MSE of prediction of a gene across all samples. A strong positive correlation or low MSE for a gene suggests high predictive power by the network on the respective gene.

## Single-cell test data processing

Human embryonic stem cell differentiation data [16] were downloaded from the Gene Expression Omnibus (GEO, accession number GSE75748) in the format of expected counts. The downloaded data were converted to RPM (reads per million). Renal cell carcinoma data [19], in the format of normalized UMI counts, and corresponding metadata, were download via the Single Cell Portal. Data was reduced to cells from patient P915. Cells with library sizes more than 3 median absolute deviations above the median were removed as potential doublets. 2000 cells were randomly selected for further analysis and underwent reversion of the log-transformation. Human embryonic kidney (HEK) cell read and UMI data, sequenced with Smart-seq3 [15], were downloaded from ArrayExpress (accession code E-MTAB-8735). Ensembl IDs were converted to gene symbol and data was normalized for library size (RPM). Oligodendroglioma data [17] were downloaded from GEO (accession number GSE70630) as $\log_2(\text{TPM}/10+1)$ and converted back to TPM. Lung Atlas 10X and Smart-seq2 data [18], together with corresponding metadata, were downloaded from Synapse (ID syn21041850). For both sequencing methods, data was restricted to cells from the lung of patient 1. Potential doublets (cells with library sizes above the median) were removed from the 10X data. For this dataset, library sizes above the median were considered potential doublets due to the bi-modal distribution of library sizes below the usual threshold of median+(3*MAD). After doublet removal, 2000 cells were randomly selected for RPM normalization and further analysis for both sequencing methods.

## Dropout imputation

Version 0.0.9 of scImpute(Li and Li 2018) was used for dropout imputation, in "TPM" mode for the oligodendroglioma dataset and "count" mode for all other datasets, without specifying cell type labels. The parameters were left as default, except for drop_thre = 0.3 (upon artificial

masking), as the default of 0.5 resulted in no imputations performed. Cell cluster number (Kcluster) was left at the default value of 2 for imputation of all datasets except for the hESC differentiation datasets (snapshot and timecourse), where it was set to 6 in order to match the number of cell clusters identified by the authors [16], and the Smart-seq3 dataset, where it was set to 1 because only one cell type was sequenced. SAVER 1.1.1 was used with size.factors = 1. SCRABBLE 0.0.1 was run with the parameters suggested by the authors and using by default the average gene expression across cells as the bulk reference. In the case of the hESC differentiation dataset, bulk data from the same study was available, and thus was also used as reference. kNN-smoothing [8] (python implementation v. 2.1) was run on the data before library size normalization, as this method performs a different correction. Since the Oligodendroglioma data was retrieved in TPM format, kNN-smoothing was not included in the method comparison for this dataset. For all other imputation methods, the data was log2-transformed with a pseudocount of 1. DrImpute 1.0 was run using the default parameters. For dropout imputation by average expression ('Baseline'), gene expression levels were log2-transformed with a pseudocount of 1 and the average expression of each gene across all cells, excluding zeros, was used for imputation. For network-based imputation, expression values were log2-transformed with a pseudocount of 1 and centered gene-wise across all cells. The original centers were stored for posterior re-conversion. Subsequently, cell-specific deviations of expression levels from those centers were predicted using either Eq (5) or the following iterative procedure. During the iteration genes were first predicted using all measured predictors. Subsequently, genes with dropout predictors were re-predicted using the imputed values from the previous iteration. This was repeated for at most 50 iterations. The obtained values were added to the gene-wise centers. We note that the values after imputation cannot be interpreted as TPMs/RPMs, as the sum of the expression levels per sample is no longer guaranteed to be the same across samples. However, one could still perform a new normalization by total signal (sum over all genes) to overcome this issue.

## Masking procedures

In order to compare the imputation error of the tested methods, we randomly masked (set to zero) some of the values for each gene, using two different approaches.

The first approach consisted of setting a fraction of the quantified, uniformly sampled values to zero for each gene (Fig 1, S4 and S5) - 35% for the hESC differentiation and Smart-seq3 datasets, 10% for the oligodendroglioma and Lung Atlas (FACS-sorted) datasets and 8% for the Renal Cell Carcinoma and Lung Atlas 10X datasets. In case of S6 Fig 30% of the cells (not genes) were sampled. This unbiased masking scheme is in agreement with previous work [35]. The differing percentages of masked values per gene in each dataset result in a comparable sparsity of the data after masking.

As an alternative masking procedure that represents more closely a downsampling process, we modelled for each gene its probability to be an observed zero in the following way: the fraction of cells where each gene was not captured (zero in the original data) was modelled as a function of its average expression across cells (S8 Fig). For this, a cubic spline was used, with knots at each 10% quantile of the average expression levels, excluding the 0% and 100% quantiles. A cubic spline was chosen so that it could properly fit to both UMI-based and non UMI-based datasets. With this model, a 'dropout probability' $p$ was computed for each gene from its mean expression. The masking procedure then consisted of, for each entry, sampling a Bernoulli distribution with probability of success $1-p$, where 0 corresponds to a mask (the entry is set to 0) and 1 to leaving the data as it is. Thus, each entry in the data matrix may be masked with a probability $p$, which is gene-specific and based on the observed dropout rates in the dataset at hand.

We observed the same relative performance of the imputation methods under this alternative masking scheme (S9 Fig), and for this reason present the results obtained with the first masking approach.

## Imputation performance analysis

Imputation was performed with each of the four tested methods separately and the imputed masked entries were then compared to the original ones. For genes where at least 10 values were imputed (non-zero after imputation, zero after masking) by all methods, the Pearson correlation between original and imputed values across cells was computed for each gene individually.

Additionally, we used the mean of the squared imputation error across all imputations for a given gene:

$$\frac{\sum (original - imputed)^2}{number\ of\ imputations} \tag{6}$$

In order to avoid higher errors for more highly expressed genes, we split the genes into expression quartiles when reporting the imputation error.

## Top performing methods

For each gene, the best performing method (highest correlation / lowest MSE) was computed. For the correlation-based analysis, methods with a correlation difference to the maximum no bigger than 0.1 were considered to be top performing. If all methods for a given gene were within this range (all top performers), the gene was not included in the foreground of Figs 2 and S10. For the MSE-based analysis, the maximum MSE difference to the best performer was computed for each gene individually, since the MSE range can be quite different from gene to gene. The maximum MSE difference was determined as 5% of the MSE range, in order to make it comparable to the threshold used in the correlation-based analysis and thus, methods in the top 5% of the gene-specific MSE range were considered to be top performing.

## Cell trajectory inference

Trajectory inference was performed on the original and imputed hESC time course dataset, using the R package *slingshot* (v. 1.8.0). Dimensionality reduction was done via PCA without scaling, as advised by the authors. The first 2 Principal Components were used, as retaining more did not change the relative performance of the methods. Since specifying the start and/or end of clusters sometimes resulted in branched trajectories, cluster labels were not given as input in order to avoid branching (branching was not expected in this dataset). The Pearson correlation between the obtained pseudotime and the known timepoint labels of the dataset was used for evaluation.

## Dimensionality reduction

Dimensionality reduction on the original hESC differentiation data (Fig 5B) was performed using ZINBWaVE, t-SNE and UMAP (S12 and S13 Figs). H1 and TB cells in Batch 3 were removed to avoid confounding batch effects and dimensionality reduction was performed for the remaining cells. UMAP was performed on the first 5 principal components obtained from the top 1000 most variable genes in the hESC differentiation data (normalized, $\log_2$-transformed) before and after imputation (S12 Fig) using the *Seurat* R package [25], version 4.0.4, with parameters *umap.method* = *"umap-learn"* and *metric* = *"correlation"*. ZINB-WaVE,

implemented in the R package *zinbwave* [30], was used to extract 2 latent variables from the information contained in the top 1000 genes with highest variance across cells. Batch information and the default intercepts were included in the ZINB-WaVE model, using *epsilon* = 1000 (S13 Fig). K-means clustering (k = 6) on the 2 latent variables strongly matched the annotated cell type labels (0.977 accuracy), confirming the reliability of this approach. t-SNE was performed on the normalized and $\log_2$-transformed data using the *Rtsne* R package with default settings (S13 Fig).

## Marker detection

Cluster-specific markers were detected from the $\log_2$-transformed normalized data using Seurat version 4.0.4. Detection rate was regressed out using the *ScaleData* function with *vars.to. regress* = *nGene*. Markers were detected with the *FindAllMarkers* function, using MAST [36] test and setting *logfc.threshold* and *min.pct* to 0, and *min.cells.gene* to 1.

## GO term enrichment and transcription factor analyses

All GO term enrichment analyses were performed with the *topGO* R package [37]. Enrichment in GO biological process terms among cluster-specific markers (Fig 4B and 4C) was performed for each cell cluster and (no) imputation method separately, using as foreground the set of significant cluster markers detected by Seurat, with FDR < 0.05 and |logFC| > 0.25, and as background all genes in the Seurat result (both significant and non-significant). The *classic* algorithm was used, in combination with Fisher test, and $\log_2$ enrichment was quantified as the $\log_2$ of the ratio between the number of significant and expected genes in each term. Significantly enriched (p-value < 0.001 and $\log_2$ enrichment > 0.5) GO biological process terms within each set of cluster markers, as detected in the original data (no masking, no imputation), were defined as "high confidence" terms.

For regulatory GO molecular function term enrichment analyses (Fig 5A), significant (FDR < 0.05 and |logFC| > 0.25) markers uniquely detected without / with each imputation method were combined across all clusters and tested for enrichment in the term "DNA-binding transcription factor activity" against the background of all genes obtained as the result of Seurat (both significant and non-significant). The *classic* algorithm was used, in combination with Fisher test, and $\log_2$ enrichment was quantified as the $\log_2$ of the ratio between the number of significant and expected genes in each term.

To identify transcription factors (TFs) among cluster markers exclusively detected using the network-based method, a curated TF list was downloaded from http://tfcheckpoint.org/.

## Determination of the optimal imputation method per gene

In order to determine the best performing imputation method for each gene, 70% of the cells in each dataset were used as training data, where a percentage of the expression values were masked, as previously described. In the Smart-seq3 dataset, where only around 100 cells were available, the training was done in 98% of the dataset The remaining cells were used for testing. After masking, each of the tested imputation methods was applied to the training data and the imputed values of masked entries were then compared to the measured values. The Pearson correlation coefficient was computed for each gene with at least 10 imputed (with a non-zero value) masked entries. For each gene, the method leading to the highest correlation coefficient was chosen as optimal. When no decision could be done, the Baseline method was used as a default.

### The ADImpute R package

The ADImpute R package is composed of two main functions, *EvaluateMethods* and *Impute*. *EvaluateMethods* determines, for each gene, the method resulting in the best imputation performance, in a cross-validation procedure. *Impute* performs dropout imputation according to the choice of method provided by the user. Currently supported methods are scImpute, DrImpute, SAVER, SCRABBLE, the Baseline and Network methods described in this manuscript and an Ensemble method, which takes the results from *EvaluateMethods* to select the imputation results from the gene-specific best method. Additionally, the user can choose to estimate the probability that each dropout value is a true zero, according to the approach used by scImpute, and leave the values unimputed if their probability of being a true zero falls above a user-defined threshold.

## Supporting information

**S1 Table. Characteristics of the test datasets.**
(XLSX)

**S2 Table. Spearman rank correlation between original values and imputation results after masking.** Correlation was computed between the vector of original entries and imputations common to all methods.
(XLSX)

**S3 Table. Percentage of masked dropouts imputed by each method in the tested datasets.**
(XLSX)

**S4 Table. Percentage of genes best imputed by each method (lowest MSE) in the tested datasets restricted to values that could be imputed by all methods.**
(XLSX)

**S1 Fig. Correlation between network predictions (using the model from Seifert et al. [14] (light purple) and the improved network described here (blue)) and measured gene expression in diverse TCGA datasets.** For each gene its expression was predicted in a given tumor sample using the measured expression values of all detected predictors in the model. Subsequently, observed and predicted values were correlated across all samples from one cohort. The plots show the distributions of Pearson's correlation scores across all genes common between the network model and the respective TCGA dataset. Although there is variation with respect to how well genes in different tumor entities can be predicted, the distributions are always strongly skewed in favour of positive correlations. This trend is enhanced with the new model presented here. AML—Acute Myeloid Leukemia; BRCA—Breast Invasive Carcinoma; COAD—Colon Adenocarcinoma; GBM—Glioblastoma Multiforme; HNSC—Head and Neck Squamous Cell Carcinoma; KIRC—Kidney Renal Clear Cell Carcinoma; LUAD—Lung Adenocarcinoma; LUSC—Lung Squamous Cell Carcinoma; OV—Ovarian Serous Cystadenocarcinoma; READ—Rectum Adenocarcinoma; SKCM—Skin Cutaneous Melanoma; STAD—Stomach Adenocarcinoma; THCA—Thyroid Carcinoma.
(PDF)

**S2 Fig. Correlation between network predictions (using the network described here–blue—, a partially randomized network—light blue—and a fully randomized network–grey) and measured gene expression in diverse healthy tissues from the GTEx consortium.** For each gene its expression was predicted in a given healthy tissue sample using the measured expression values of all detected predictors in the model. Subsequently, observed and predicted values were correlated across all samples from one tissue. The plots show the distributions of

Pearson's correlation scores across all genes common between the network model and the GTEx data.
(PDF)

**S3 Fig. Comparison between results of the network-based imputation using the iterative approach and the Moore-Penrose pseudoinversion (MP) with varying tolerance thresholds in a random subset of 20 cells from the hESC differentiation dataset.** A) Correlation between the results of the iterative approach (x axis) and the Moore-Penrose pseudoinversion (y-axis), across the 20 random cells. B) Imputation performance per gene using the iterative approach and MP with different tolerance thresholds (Pearson correlation, top, and MSE, bottom), separated by expression quartile on the masked data. The higher the tolerance threshold, the fewer singular values are used for the pseudoinversion. Results were limited to imputations performed by all methods.
(PDF)

**S4 Fig. Imputation performance of Network (blue), DrImpute (green), kNN-smoothing (pink), SAVER (yellow), scImpute (turquoise), SCRABBLE (with bulk or single-cell data as reference; purple) and Ensemble (orange), stratified by expression quartiles, on additional datasets. A)** Distribution of average expression levels in each dataset. Quartiles are represented by vertical lines. **B)** Pearson correlation coefficient, for each gene, between the imputation by the specified method and the original values before masking. Only values that could be imputed by all methods were used for correlation computation. Expression quartiles are determined for each dataset separately, on the masked data.
(PDF)

**S5 Fig. Imputation error of Baseline (slate grey), Network (blue), DrImpute (green), kNN-smoothing (pink), SAVER (yellow), scImpute (turquoise), SCRABBLE (with bulk or single-cell data as reference; purple) and Ensemble (orange), stratified by expression quartiles.** Only values that could be imputed by all methods were used for MSE computation. Expression quartiles are determined for each dataset separately, on the masked data. The x axis is presented log-transformed and was cropped at 0.1 to exclude the low-MSE tail from visualization and facilitate result comparison.
(PDF)

**S6 Fig. Imputation performance of Baseline (slate grey), Network (blue), DrImpute (green), kNN-smoothing (pink), SAVER (yellow), scImpute (turquoise) and SCRABBLE (purple) methods in the hESC differentiation dataset, upon cellwise masking.** Correlation coefficient between imputed and original values (top) and MSE of imputation (bottom), upon random masking of 30% of the quantified genes in each cell in the hESC differentiation dataset. The Baseline method computes the average expression of a gene across all cells and it is not using the information of any other genes. Hence, one would assume that its error should be independent of the number of missing genes per cell. This is however not the case due to a biased gene sampling: cells with few detected genes will preferentially report values for highly expressed genes, whereas cells with many detected genes will represent a less biased sample of the whole transcriptome. This is affecting the performance, which thus is slightly dependent on the number of missing genes per cell. Values were restricted to imputations performed by all tested methods.
(PDF)

**S7 Fig. Imputation performance of the Network method upon randomization in the hESC differentiation dataset.** Dropout imputation was performed using the network described here

(blue), a partially randomized network (light blue) and a fully randomized network (grey).
(PDF)

**S8 Fig. Comparison of the two masking procedures employed in this work: a uniform masking procedure which sets to zero the same number of entries per gene (UniformMask) and a model-based procedure which sets entries to zero with a gene-specific probability obtained from the data.** Comparisons were done on representative datasets of non-UMI data (**A**), **C**), **E**), **G**)) and UMI data (**B**), **D**), **F**), **H**)). **A)** and **B)** Fit of the spline model (Methods) to the original data. **C)** and **D)** Fraction of zeros in the data before (Original) and after (Uniform-Mask, ModelMask) masking, compared to original average gene expression,. **E)** and **F)** Distribution of mean expression and expression variance before and after masking, compared to original average gene expression. **G)** and **H)** Mean-variance trend before and after masking.
(PDF)

**S9 Fig. Imputation performance of Baseline (slate grey), Network (blue), DrImpute (green), SAVER (yellow), scImpute (turquoise) and SCRABBLE (purple) methods in the hESC differentiation dataset, using a gene-specific masking procedure (Methods).** Correlation coefficient between imputed and original values (top) and MSE of imputation (bottom), upon random masking of 30% of the quantified genes in each cell in the hESC differentiation dataset. Only values that could be imputed by all methods were used for performance analysis. Expression quartiles are determined on the masked data. The MSE axis is presented log-transformed and was cropped at 0.1 to exclude the low-MSE tail from visualization and facilitate result comparison. kNN-smoothing is not included in the comparison since no imputations could be performed by this method.
(PDF)

**S10 Fig. Characterization of the genes best predicted by Network (blue), DrImpute (green), kNN-smoothing (pink), SCRABBLE (single cell and bulk data as reference; purple), scImpute (turquoise) and SAVER (yellow) in the hESC differentiation dataset.** Distribution of missing values per gene, average expression levels and variance of the genes for which a given method is one of the top performers, compared against all tested genes (background). Average gene expression is shown as $\log_2$-transformed normalized expression. Methods with close performance to the best method (correlation not smaller than 0.1 –best method) are selected as top best performing. Genes for which all methods are best performers are included in the background but not in the foreground.
(PDF)

**S11 Fig. Characterization of the genes best predicted by Network (blue), DrImpute (green), SCRABBLE (single cell and bulk data as reference; purple) and scImpute (turquoise) in the hESC differentiation dataset, using imputation error to quantify performance.** Distribution of missing values per gene, average expression levels and variance of the genes for which a given method is one of the top performers, compared against all tested genes (background). Average gene expression is shown as $\log_2$-transformed normalized expression. Methods with close performance to the best method (MSE not higher than 1/20 of the MSE range for that given gene) are selected as top best performing. Genes for which all methods are best performers are included in the background but not in the foreground.
(PDF)

**S12 Fig. Effect of imputation with Baseline, DrImpute, kNN-smoothing, SAVER, scImpute, SCRABBLE and Network methods on UMAP plots.** Data was subject to: no masking (left column), relaxed masking (35% of quantified entries per gene were set to zero, middle

column), stringent masking (60% of quantified entries set to zero, right column). The plot in the upper left reflects the clustering on the original, unchanged data. Imputation was performed for actually missing values in the original data (all columns) and on masked values (columns 2 & 3). Colors represent cell type label annotations from the original publication. DEC: definitive endoderm cells; EC: endothelial cells; H9: undifferentiated human embryonic stem cells; HFF: human foreskin fibroblasts; NPC: neural progenitor cells; TB: trophoblast-like cells.
(PDF)

**S13 Fig. Dimensionality reduction results with different techniques: ZINBWaVe and t-SNE on the hESC dataset.** Data was not subject to any masking. Colors represent cell type label annotations from the original publication. DEC: definitive endoderm cells; EC: endothelial cells; H9: undifferentiated human embryonic stem cells; HFF: human foreskin fibroblasts; NPC: neural progenitor cells; TB: trophoblast-like cells.
(PDF)

**S14 Fig. EHF and OSR1 expression levels before and after dropout imputation, across cell types.** DEC: definitive endoderm cells; EC: endothelial cells; H9: undifferentiated human embryonic stem cells; HFF: human foreskin fibroblasts; NPC: neural progenitor cells; TB: trophoblast-like cells.
(PDF)

**S15 Fig. PRRX1 and TWIST2 expression levels before dropout imputation, across cell types.** DEC: definitive endoderm cells; EC: endothelial cells; H9: undifferentiated human embryonic stem cells; HFF: human foreskin fibroblasts; NPC: neural progenitor cells; TB: trophoblast-like cells.
(PDF)

## Acknowledgments

The results here shown are in part based upon data generated by the TCGA Research Network: http://cancergenome.nih.gov/. The Genotype-Tissue Expression (GTEx) Project was supported by the Common Fund of the Office of the Director of the National Institutes of Health, and by NCI, NHGRI, NHLBI, NIDA, NIMH, and NINDS. The data used for the analyses described in this manuscript were obtained from the GTEx Portal on 12/2019.

We gratefully acknowledge help from Dr. Michael Seifert (TU Dresden, Germany) on the construction of the transcriptional regulatory network.

## Author Contributions

**Conceptualization:** Ana Carolina Leote, Andreas Beyer.

**Data curation:** Xiaohui Wu.

**Funding acquisition:** Andreas Beyer.

**Investigation:** Ana Carolina Leote, Andreas Beyer.

**Methodology:** Ana Carolina Leote, Xiaohui Wu, Andreas Beyer.

**Project administration:** Andreas Beyer.

**Resources:** Andreas Beyer.

**Software:** Ana Carolina Leote, Xiaohui Wu.

**Supervision:** Andreas Beyer.

**Visualization:** Ana Carolina Leote.

**Writing – original draft:** Ana Carolina Leote, Xiaohui Wu, Andreas Beyer.

**Writing – review & editing:** Ana Carolina Leote, Andreas Beyer.

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
