## [Decision Letter · Decision Letter 0]

18 Sep 2021

Dear Dr. Beyer,

Thank you very much for submitting your manuscript "Manuscript “Regulatory network-based imputation of dropouts in single-cell RNA-sequencing data”" for consideration at PLOS Computational Biology.

As with all papers reviewed by the journal, your manuscript was reviewed by members of the editorial board and by several independent reviewers. In light of the reviews (below this email), we would like to invite the resubmission of a significantly-revised version that takes into account the reviewers' comments.

Please revise it according to the reviewers carefully.

We cannot make any decision about publication until we have seen the revised manuscript and your response to the reviewers' comments. Your revised manuscript is also likely to be sent to reviewers for further evaluation.

Sincerely,

Quan Zou

Guest Editor

PLOS Computational Biology

Ilya Ioshikhes

Deputy Editor

PLOS Computational Biology

Please revise it according to the reviewers carefully.

Reviewer's Responses to Questions

**Comments to the Authors:**

Reviewer #1: In this manuscript, the authors proposed a transcriptional regulatory network learned from bulk gene information for single-cell RNA sequencing data imputation. The network approach showed good performance for lowly expressed genes, which could be utilized for the imputation of markers and regulators. Additionally, the authors make a point that there is no best imputation approach but only the best method for a particular combination for different circumstances. Overall, the manuscript is well organized. I have some suggestions for the authors to improve their work.

1. The authors argued that the “Baseline” method, which simply calculated the average expression value of the gene, performed well for the imputation. It is astonished that such a simple method achieved such good results. I believe that this simple method must have been reported and compared before in single-cell imputation field. I would suggest the authors to find related papers and add more discussion to this finding.

2. Hou et al. have made a terrific benchmark work for 18 single-cell RNA sequencing imputation methods:

Hou, W., Ji, Z., Ji, H. et al. A systematic evaluation of single-cell RNA-sequencing imputation methods. Genome Biol 21, 218 (2020).

Based on their conclusion an important state-of-the-art method, kNN-smoothing, was missing in this manuscript. I suggest the authors to add this method in this work.

3. The URL “http://www.tfcheckpoint.org/index.php/browse” in line 581 is not accessible, please check the link.

4. More details of previously learnt regulatory models was encouraged to be added.

5. Several abbreviations don’t have their full names for their first appearance, for instance, “MAGIC“ in line 77.

Reviewer #2: The paper aims to improve the performance of imputation on scRNA-Seq data through a transcriptional regulatory network learned from external, independent gene expression data. This study implemented an R-package called ADImpute that automatically determines the best imputation method for each gene in a dataset. The methods and results reported in this paper demonstrate that their network-based approach outperforms published state-of-the-art methods. The work proposed that imputation should maximally exploit external information and be adapted to gene-specific features, such as expression level and expression variation across cells.

This manuscript is a reader-friendly and well-written manuscript. The following concerns need to be addressed.

Major issues:

1. As far as I know, the zero values in scRNA-Seq data are partly due to dropout events and partly due to gene expression. How can the authors ensure that the method proposed to impute the dropout zeros rather than incorrectly filling gene expression?

2. The challenges of improving the imputation of scRNA-Seq should be explained in-depth, such as what are the main challenges in the study of imputation methods and how these studies do may overcome these challenges.

3. The proposed model was compared with several methods, but lack of discussion of details regarding the weaknesses. Could the authors give more analysis about why the proposed model does better than previous work? More insightful discussions should be given.

Reviewer #3: In this paper, Leote et al. proposed a regulatory network-based imputation method for dropouts in scRNA-seq data. I have the following concerns regarding this manuscript.

1. For the evaluation of imputation methods, although the authors used synthetic data and simulation, their real data analysis was rather limited. The only real data results are in the section “Network-based imputation uncovers cluster markers and regulators.” However, other commonly used downstream analyses, including clustering, DE gene analysis, and cell trajectory inference, were not evaluated on the imputed data. The authors should add more real data results to demonstrate that their imputation methods can benefit downstream analyses better than existing imputation methods do.

2. If authors want to claim that their imputation method can benefit cell marker detection, the first step should be to find meaningful cell clusters and label them as cell types. However, the cell clustering results were not shown.

3. I do not find the current real data results convincing. For example, in Figure 3A, there is little difference between the cluster markers found on the imputed data (by the proposed method) and on the original data. Hence, this figure did not make a strong point.

4. In Figure 1, can the author label the sequencing technology for the hESC data as well? It seems that the proposed network imputation favors the UMI data. If this is the case, please clarify it.

5. For Figure 2, a distance metric should be used to quantify the similarities or differences between the background and the imputed data.

6. For Figure 3, the labels for different imputation methods are not in the same order in the four panels, making it difficult for a direct comparison. Why was the original data omitted in panel B?

7. In line 465, why does a high MSE suggest better results?

**Have the authors made all data and (if applicable) computational code underlying the findings in their manuscript fully available?**

Reviewer #1: Yes

Reviewer #2: Yes

Reviewer #3: None

PLOS authors have the option to publish the peer review history of their article (what does this mean?). If published, this will include your full peer review and any attached files.

Reviewer #1: No

Reviewer #2: No

Reviewer #3: No
---

## [Decision Letter · Decision Letter 1]

27 Dec 2021

Dear Dr. Beyer,

Thank you very much for submitting your manuscript "Regulatory network-based imputation of dropouts in single-cell RNA sequencing data" for consideration at PLOS Computational Biology. As with all papers reviewed by the journal, your manuscript was reviewed by members of the editorial board and by several independent reviewers. The reviewers appreciated the attention to an important topic. Based on the reviews, we are likely to accept this manuscript for publication, providing that you modify the manuscript according to the review recommendations.

Sincerely,

Quan Zou

Guest Editor

PLOS Computational Biology

Ilya Ioshikhes

Deputy Editor

PLOS Computational Biology

[LINK]

Reviewer's Responses to Questions

**Comments to the Authors:**

Reviewer #1: I am satisfied with the answers the authors provided to my questions. As for the newly added part of the authors (lines 73-82), I would suggest the authors to add another citation concerning technical dropout:

Zhang, Z., Cui, F., Wang, C., Zhao, L. and Zou, Q. (2021) Goals and approaches for each processing step for single-cell RNA sequencing data. Briefings in bioinformatics, 22.

Reviewer #2: The authors has answered all my questions.I have no more questions.

**Have the authors made all data and (if applicable) computational code underlying the findings in their manuscript fully available?**

Reviewer #1: Yes

Reviewer #2: Yes

PLOS authors have the option to publish the peer review history of their article (what does this mean?). If published, this will include your full peer review and any attached files.

Reviewer #1: No

Reviewer #2: No

Figure Files:

Data Requirements:

Reproducibility:

References:

---

## [Editor Report · Decision Letter 2]

18 Jan 2022

Dear Dr. Beyer,

We are pleased to inform you that your manuscript 'Regulatory network-based imputation of dropouts in single-cell RNA sequencing data' has been provisionally accepted for publication in PLOS Computational Biology.

Best regards,

Quan Zou

Guest Editor

PLOS Computational Biology

Ilya Ioshikhes

Deputy Editor

PLOS Computational Biology

---

## [Editor Report · Acceptance letter]

14 Feb 2022

PCOMPBIOL-D-21-01399R2 

Regulatory network-based imputation of dropouts in single-cell RNA sequencing data

Dear Dr Beyer,

I am pleased to inform you that your manuscript has been formally accepted for publication in PLOS Computational Biology. Your manuscript is now with our production department and you will be notified of the publication date in due course.

With kind regards,

Katalin Szabo
